# Effect of Different Bonding Materials on Flip-Chip LED Filament Properties

**Chengyu Guan [1], Jun Zou [1,2,\*], Qingchang Chen [1,3,\*], Mingming Shi [1] and Bobo Yang [1]**

[1] School of Science, Shanghai Institute of Technology, Shanghai 201418, China; 186181302@mail.sit.edu.cn (C.G.); mmshi@sit.edu.cn (M.S.); boboyang@sit.edu.cn (B.Y.)
[2] Institute of New Materials and Industrial Technology, Wenzhou University, Wenzhou 325024, China
[3] School of Urban Construction and Safety Engineering, Shanghai Institute of Technology, Shanghai 201418, China
[\*] Correspondence: zoujun@sit.edu.cn (J.Z.); cqc@sit.edu.cn (Q.C.)

**Abstract:** This article researches the effect of Sn-based solder alloys on flip-chip light-emitting diode LED (FC-LED) filament properties. SEM images, shearing force, steady-state voltage, blue light luminous flux, and junction temperature were examined to demonstrate the difference between two types of FC-LED filaments welded with two solders. The microstructure surface of Sn90Sb10 filament solder joints was smoother and had fewer voids and cracks compared with that of SAC0307 filament solder joints, which indicates that the Sn90Sb10 filaments had a higher shearing force than the SAC0307 filaments; moreover, the average shearing force was beyond 200 gf (standard shearing force). The steady-state voltage and junction temperature of the Sn90Sb10 solder-welded FC-LED filament were lower, and the Sn90Sb10 filament had a relatively higher blue light luminous flux. If high reliability of the solder joints and better photoelectric properties of the filaments are required, this Sn90Sb10 solder is the best bonding material for FC-LED filament welding.

**Keywords:** FC-LED filament; lead-free solder; photoelectric properties; reliability

## 1. Introduction

As a new generation of light source, the light-emitting diode (LED) has many advantages, such as energy saving, environmental protection, and no radiation emission [1–3]. With the growing demand of LEDs, the white LED has been industrialized [4,5]. Flip-chip LED (FC-LED) filament bulbs are gradually replacing incandescent lamps, with the advantages of lower working voltage, less power consumption, and higher luminous efficiency. Due to the output power of LED chips being improved, the FC-LED filament package process requires advanced technology [6–9].

Compared with the traditional LED packaging (wire-bonded LED) process, the FC-LED package process has improved the bonding efficiency and thermal conductivity of the FC-LED filament. Industrial FC-LED chip-bonding technologies include conductive adhesive bonding and eutectic welding [10], of which, the FC-LED filament that is bonded by eutectic welding has high reliability. In addition, in the FC-LED packaging process, alloys that have properties such as good wettability and low cost can be used for eutectic welding [11]. The improved properties of the FC-LED packaging structure, such as good heat dissipation and low stress, make it an essential technology. Lead-free solder alloys are gradually replacing conductive silver glue and thermal conductivity adhesive, due to the advantage of high heat dissipation efficiency in the LED wafer package field. In order to improve the properties of the FC-LED filament, Sn90Sb10 (Sn90–Sb10) and SAC0307 (Sn–0.3Ag-0.7Cu) solders were selected for the bonding process. Sn90Sb10 belongs to the Sn-based solder alloy, which also can be used for bonding the flip-chip LED [12,13]. At present, SAC series solders have been widely used

in the production of industrial FC-LED filaments (their main components are Sn/Ag/Cu and flux). Our aim was to find new types of Sn-based solders with better performance [14–16]. In this study, Sn90Sb10 solder was used in order to study ways to improve the reliability and the photoelectric properties of the FC-LED filament.

## 2. Experimental Section

A flip chip (Zhongke0620, Shenzhen, China) with an Au pad was welded to a flexible aluminum substrate. The flip chip size was 6 mil × 20 mil, the working voltage was 2.51–2.60 V, and the mainly blue wavelength range was 452.5 nm to 454.9 nm. In this research, JUST BT-929 (JUST, Xiamen, China) LED automatic bonding equipment was used for attaching the flip chip to the substrate; Sn90Sb10 (Proudbull, Shenzhen, China) and SAC0307 (Proudbull, Shenzhen, China) solders were used as bonding materials. A flexible aluminum substrate was placed on a horizontal movable platform, and the flip chip was fixed on the wafer substrate; then, the wafer substrate was removed. All samples were welded using the reflow process. The physical morphology of the flip chip is presented in Figure 1; Figure 2 shows the welding schematic diagram, composed of, from top to bottom, the flip chip, the lead-free solder, an Au layer, a Cu layer, and the flexible aluminum substrate.

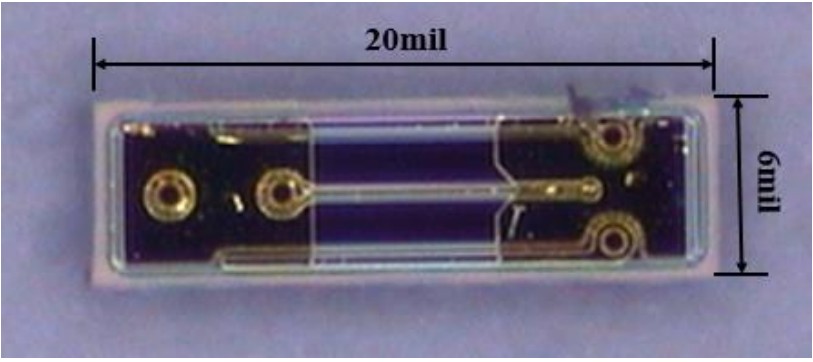

**Figure 1.** Physical morphology of the flip-chip.

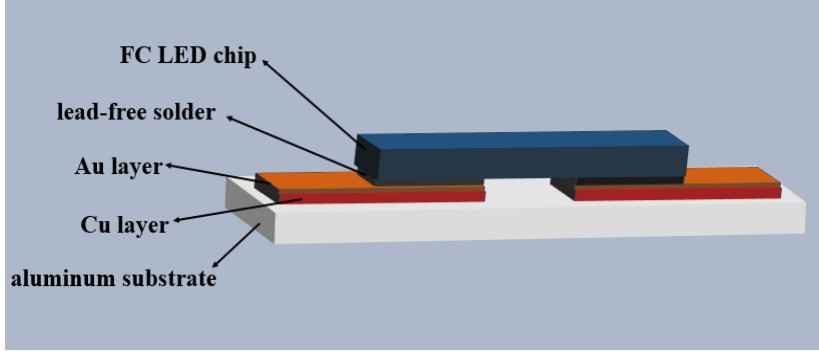

**Figure 2.** Flip chip welding schematic diagram. FC-LED: flip-chip light-emitting diode.

Figure 3a shows both the SAC0307- and the Sn90Sb10-welded filaments, with SAC0307 filaments on the left, and Sn90Sb10 filaments on the right. Figure 3b,c show the two types of filament with input current. The substrate for these types of filament was 175 mm thick, the working voltage was about 265 V, and each filament could fix 100 chips in total. A thermocouple was attached to the substrate for thermal performance analysis in a reflow welding test. The peak welding temperature was measured by the LED-T300B thermal analysis system (LEETS, Shanghai, China). It was found in the reflow welding test that the performance of the filaments welded at the peak temperature range was better. More precisely, the peak welding temperature of SAC0307 was about 553.5 K, and its melting point was

about 500.5 K. The melting point of Sn90Sb10 was 513.5–518.5 K, and the peak welding temperature of the solder was about 573.5 K [17,18].

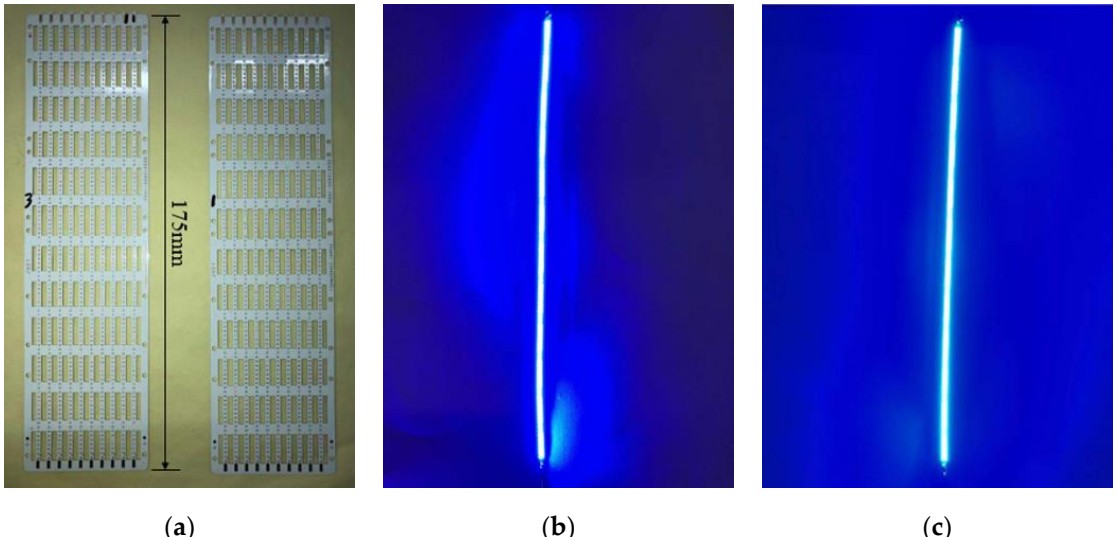

| **(a)** | **(b)** | **(c)** |

**Figure 3.** (**a**) Sn–0.3Ag–0.7Cu (SAC0307) and Sn90–Sb10 (Sn90Sb10) solder-welded substrate, (**b**) SAC0307 filament and (**c**) Sn90Sb10 filament with current.

The physical morphology of this flip chip was observed by two-dimension analysis. The microstructure surface of the solder joints after the shear test were photographed by SEM (Hitachi, Tokyo, Japan), with a field-emission scanning electron microscope at a working voltage of 15.0 kV and a working distance of 6 mm during the test. The shearing force of the two types of solder joint was examined using a shearing strength tester (XYZETC, Shenzhen, China). In the shearing test, the instrument working at room temperature provided a thrust of 200 Kgf and detached the flip chip from the bonded solder joint. A LED-T300B thermal analysis system (LEETS, Shanghai, China) was used for measuring the junction temperature of the FC-LED filament enclosed in the bulb at room temperature, at a test voltage of 285 V with input current of 15 mA. A GDJS-100 (Aoke, Hangzhou, China) high-temperature and high-humidity aging equipment was used for the FC-LED filament aging test at 85 °C/85% relative humidity (RH). The steady-state voltage of these FC-LED filaments was measured by a constant current source of 285 V/15 mA, lighted for 30 min at room temperature. The blue light luminous flux of these filaments was calculated by the LED300 integrating sphere (EVERFINE, Hangzhou, China).

## 3. Results and Discussion

### 3.1. Microstructure of the Solder Joint after the Shear Test

Figure 4 shows the microstructure images of the two types of filament solder joints after the shear test. As shown in Figure 4a,b, the microstructure surface of the Sn90Sb10 filaments' solder joints is smoother than that of the SAC0307 filaments' solder joints. More voids and cracks are present in the solder joints of the SAC0307 filaments, and as evident in Figure 4a, cracks are most likely to occur where voids are [19–21].

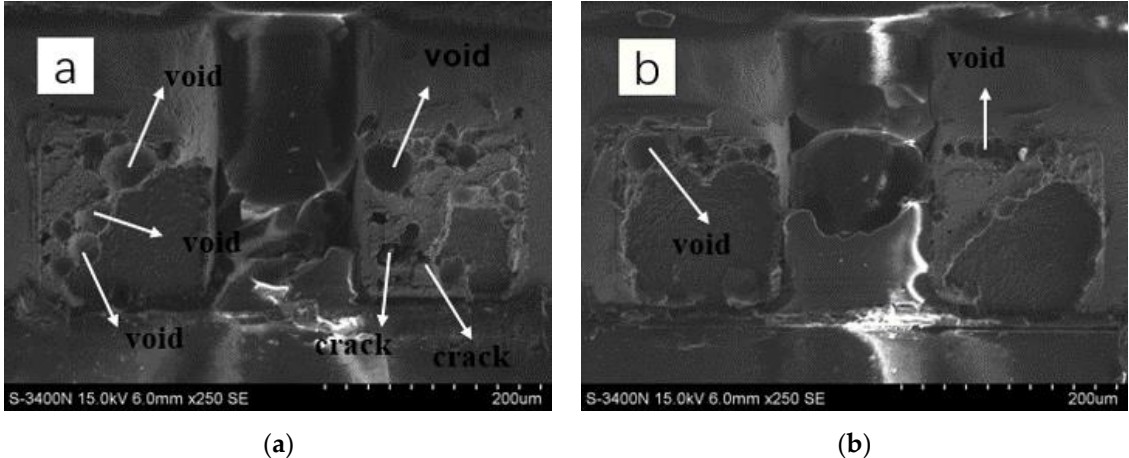

(**a**)　　　　　　　　　　　　　　　　　　　　　　　(**b**)

**Figure 4.** Microstructure of the two types of solder joint: (**a**) SAC0307 and (**b**) Sn90Sb10.

*3.2. Shearing Force of the Solder Joint*

Figure 5 shows the shearing force for FC-LED filament solder joints obtained with SAC0307 and Sn90Sb10 solders. The shearing force fir both solder joints was over 200 gf (standard shearing force). Figure 5 shows that the average shearing force for the Sn90Sb10 solder joint was higher than that for the SAC0307 solder joint. The average shearing force for the SAC0307 filament solder joints was approximately 305.25 gf, and that for the Sn90Sb10 filament solder joints was approximately 340.35 gf; therefore, that for the Sn90Sb10 filaments was 11.5% higher than that for the SAC0307 filaments. We also found that the lower quartile of the SAC0307 shearing force was 268 gf, and the upper quartile was 337.5 gf. The lower quartile of the Sn90Sb10 filament shearing force was 313 gf, and the upper quartile was 380.5 gf, while the median linear shearing forces of SAC0307- and Sn90Sb10-welded solder joints were 313 gf and 339.5 gf, respectively. The shearing force of a solder joint can be calculated by the formula:

$$\delta = \sqrt{\frac{\sum (X - \mu)^2}{N}}$$

where $\delta$ is the variance, $X$ is the shearing force of the sample, $\mu$ is the average shearing force, and $N$ is the number of solder joints. The shearing force variance value $\delta$ of the SAC0307 sample was calculated to be 49.99, and that of the Sn90Sb10 sample was 41.67. These data revealed that the shearing force of the Sn90Sb10 filament solder joint was more concentrated and superior to that of the SAC0307 filaments.

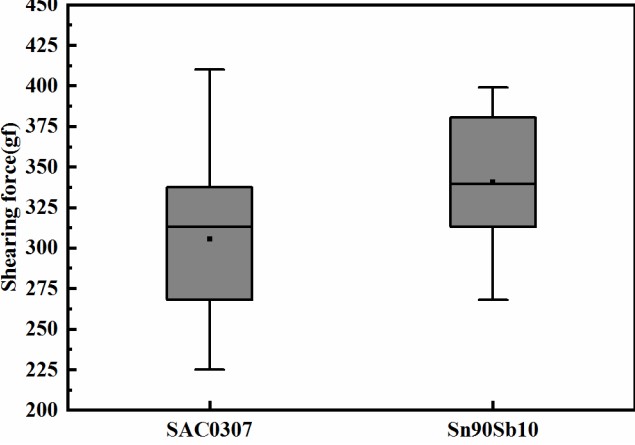

**Figure 5.** Shearing force of the two types of solder joints.

Compared with SAC0307, Sn90Sb10 has higher bonding strength, and the melting point of Sn90Sb10 is 10 °C higher than that of SAC0307. During the flip-chip bonding process, the wettability and spread of the SAC0307 solder are worse than those of the Sn90Sb10 solder [22,23]. In the eutectic welding process, the thermal fatigue resistance of the SAC0307 solder is poorer than that of the Sn90Sb10 solder, which adversely affects the strength of the solder joint. In addition, there are more voids in the SAC0307 solder joints, which leads to more cracks. It can be concluded that the Sn90Sb10 filament solder joint has fewer voids and cracks, which improved the reliability of the Sn90Sb10 filament solder joints [24–26].

### 3.3. Steady-State Voltage of the FC-LED Filament

Figure 6 shows the steady-state voltage of the two types of FC-LED filaments lit for 15 min. It can be clearly seen that the SAC0307 filaments' voltage is higher than that of the Sn90Sb10 filaments. Detailed data showed that the average steady-state voltage of the SAC0307 filaments was 265.18 V, and that of the Sn90Sb10 filaments was 263.62 V. According to these calculations, we concluded that the steady-state voltage of the Sn90Sb10 filaments was 0.59% lower than that of the SAC0307 filaments; the variance of the Sn90Sb10 filaments was 0.239, while that of the SAC0307 filaments was 0.262. A higher filament voltage will accelerate the aging rate and reduce the service life. Compared with the Sn90Sb10 filament, more voids and cracks were present in the SAC0307 filament solder joints; also, the solder paste of the Sn90Sb10 solder joints had a large contact area with the FC-LED, anf the thermal resistance and steady-state voltage were smaller [27–29].

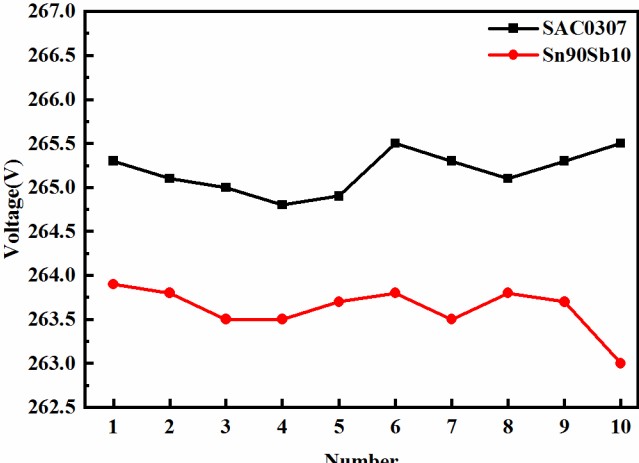

**Figure 6.** Steady-state voltage of the two types of FC-LED filaments.

### 3.4. Junction Temperature of the FC-LED Filaments

Table 1 shows the junction temperature of SAC0307 and Sn90Sb10 FC-LED filaments. The average junction temperature of the SAC0307 filaments was 123.9 °C, and that of the Sn90Sb10 filaments was 117.12 °C, thus, the average junction temperature of the Sn90Sb10 filaments was 5.47% lower than that of the SAC0307 ones. The junction temperature variance value of the Sn90Sb10 filaments was 0.249, and that of the SAC0307 filaments was 1.096, respectively, when using constant current source, while the steady-state voltage increased as the resistance increased. The semiconductor device with higher thermal resistance will have poorer thermal conductivity and worse heat dissipation performance. The junction temperature can be calculated by the thermal resistance formula:

$$Rth = \frac{L}{K \bullet S}$$

where *L* is the path length of heat conduction of the solder joint, *K* is the thermal conductivity of the lead-free solder, and *S* is the cross-sectional area of heat conduction of the solder joint.

Due to more cracks present in the solder joints of the SAC0307 filaments, the thermal resistance of the SAC0307 filaments is higher than that of the Sn90Sb10 ones. Compared with the Sn90Sb10 filaments, the SAC0307 filaments had higher steady-state voltage, which negatively affected the life time and reliability of the FC-LED filaments [30].

**Table 1.** Junction temperature of the two types of FC-LED filaments.

| NUMBER | 1 | 2 | 3 | 4 | 5 | AVG |
|---|---|---|---|---|---|---|
| **Junction Temperature of SAC0307 Filament (°C)** | 122.16 | 123.85 | 124.22 | 124.09 | 125.18 | 123.9 |
| **Junction Temperature of SN90SB10 Filament (°C)** | 116.76 | 117.3 | 116.93 | 117.27 | 117.34 | 117.12 |

### 3.5. Optical Performance of the FC-LED Filaments

Figure 7 shows the blue light luminous flux of the two types of FC-LED filaments lit for 30 min. The blue light luminous flux of the Sn90Sb10 filaments was higher than that of the SAC0307 filaments. The average blue light luminous flux of the Sn90Sb10 filaments was 80.81 lm, and that of the SAC0307 filaments was 76.43 lm. The blue light luminous flux variance value of the Sn90Sb10 filaments was 0.814 and that of the SAC filaments was 0.252. The Sn90Sb10 filaments' blue light luminous flux was 5.7% higher than that of the SAC0307 filaments. Figure 6 demonstrates that the steady-state voltage of the Sn90Sb10 filaments was higher than that of the Sn90Sb10 filaments, and the FC-LED filaments with higher steady-state voltage had lower blue light luminous flux and higher thermal resistance [31]. With higher thermal resistance, the current density of the FC-LED filament decreased, which directly led to the decrease of the luminous flux of the filament. In conclusion, the Sn90Sb10 filaments had longer life, higher reliability, and better photoelectric performance than the SAC0307 filaments, with the advantage of higher blue light luminous flux, lower steady-state voltage, and higher shearing strength [32,33].

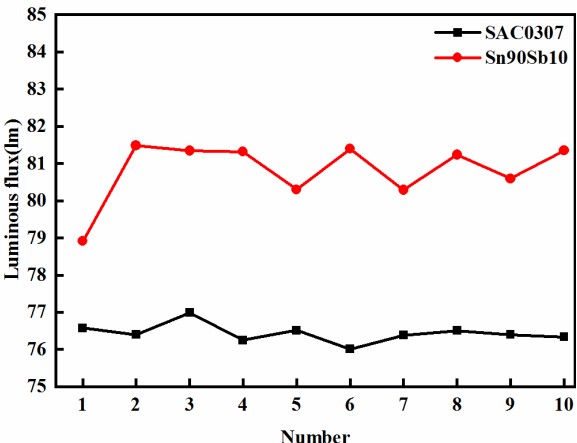

**Figure 7.** Blue light luminous flux of the two types of FC-LED filaments.

### 3.6. Photoelectric Properties of the FC-LED Filaments after Aging

The steady-state voltage of the two types of FC-LED filaments are presented in Figure 8. As shien in Figure 8, at high-temperature and high-humidity (85 °C/85% RH) aging conditions, the steady-state voltage of the Sn90Sb10 filaments decreased before aging for 100 h and then increased. Moreover, after aging for 300 h, the steady-state voltage of the SAC0307 filaments was decreased by 0.23%, and that of the Sn90Sb10 filament was decreased by 0.34% compared to the initial values. During aging of the filament, in the initial stage of aging, the test environment was at room temperature. When changing the high temperature and high humidity of the aging environment to normal temperature, an effect equivalent to the annealing effect was observed, and the defects in the solder layer were repaired to a certain extent. On the whole, the resistance of the filaments was reduced, and the steady-state voltage

decreased. However, as the aging time increased, the higher temperature destroyed the surface of the flip chip, causing the ohmic contact between the chip and the electrode to degrade. The internal delamination of the flip chip and the solder layer intensified, which caused increased resistance and poor heat conductivity. The steady-state voltage of the filaments rose, and the overall stability of the filaments was reduced, but the deterioration of the material properties inside the chip caused the blue light luminous flux to decrease [34,35].

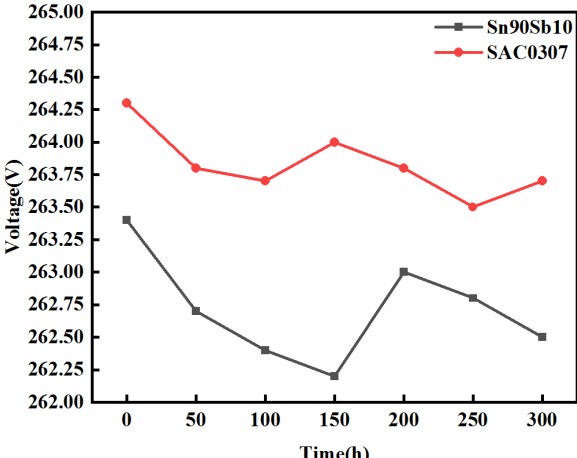

**Figure 8.** Steady-state voltage after high-temperature and high-humidity aging.

The blue light luminous fluxed of the two types of filaments are presented in Figure 9. It can be observed that the blue light luminous flux of both the Sn90Sb10 and the SAC0307 filaments decreased after aging for 300 h. Compared to the initial values, the blue light luminous flux of the Sn90Sb10 filaments was decreased by 7.82%, and that of the SAC0307 filament was decreased by 9.27%. The factors which accelerates FC-LED filaments' aging were high temperature and high humidity. The aging mechanism of the two types of filaments consisted in the acceleration of chemical reactions at high temperatures and in the diffusion of moisture in the FC-LED filament caused by high humidity, which accelerated the aging of the filaments. In addition, aging can change the material performance, cause corrosion and failure of the chip, and decrease the blue light luminous flux of the filament. In conclusion, the Sn90Sb10 filaments performed better in maintaining the blue light luminous flux; therefore, high temperature and high humidity had a greater influence on the SAC0307-welded filaments than on the Sn90Sb10-welded ones [36,37].

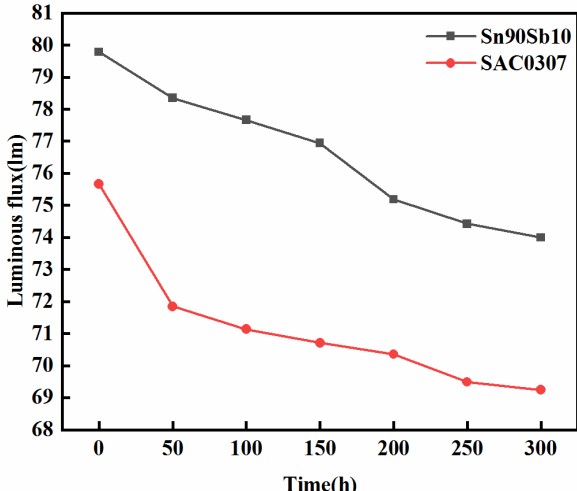

**Figure 9.** Blue light luminous flux after high-temperature and high-humidity aging.

## 4. Conclusions

In this paper, we explored the effect of different bonding materials on the properties of FC-LED filaments. The microstructure surface of Sn90Sb10 filament solder joints in failure presented fewer voids and was smoother than that of the SAC0307 filament solder joints. The average shearing force of the Sn90Sb10 FC-LED filament solder joints was 340.35 gf, 11.5% higher than that of the SAC0307 filaments solder joints of 305.25 gf, and the junction temperature of the Sn90Sb10 filaments was 117.12 °C, i.e., 5.47% lower than that of the SAC0307 filaments, amounting to 123.9 °C. Furthermore, the average voltage of the Sn90Sb10 FC-LED filaments was 263.62 V, 0.59% lower than that of the SAC0307 filaments of 265.18 V, while the Sn90Sb10 filaments had a higher blue light luminous flux of 80.81 lm, 5.7% higher than that of the SAC0307 filaments of 76.43 lm. After high-temperature and high-humidity aging, the steady-state voltages and blue light luminous fluxes of these two types of FC-LED filaments decreased. In conclusion, considering that the reliability and photoelectric performance of the Sn90Sb10-welded FC-LED filaments were relatively better with respect to those of the SAC0307 filaments, the Sn90Sb10 solder would be a better bonding material for the FC-LED welding process.

**Author Contributions:** Conceptualization, J.Z.; Methodology, J.Z.; Software, C.G.; Validation, C.G., Q.C. and B.Y.; Formal Analysis, C.G.; Investigation, C.G.; Resources, J.Z.; Data Curation, C.G.; Writing-Original Draft Preparation, C.G.; Writing-Review and Editing, C.G.; Visualization, M.S.; Supervision, Q.C. and M.S.; Project Administration, B.Y.; Funding Acquisition, J.Z. All authors have read and agreed to the published version of the manuscript.

**Funding:** This research was funded by [the Science and Technology Planning Project of Zhejiang Province, China] grant number [2018C01046], [Enterprise-funded Latitudinal Research Projects] grant number [J2016-141, J2017-171, J2017-293, J2017-243].

**Acknowledgments:** This work was supported by the Science and Technology Planning Project of Zhejiang Province, China(2018C01046), Enterprise-funded Latitudinal Research Projects(J2016-141), (J2017-171), (J2017-293), (J2017-243).

**Conflicts of Interest:** The authors declare no conflicts of interest.

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
