# Peer review of "Effect of Different Bonding Materials on Flip-Chip LED Filament Properties"

_applsci, doi:10.3390/app10010047_

Round 1

Reviewer 1 Report

The article is interesting and well thought out, however in Figure 3 the descriptions are illegible (please change the color of the descriptions). In addition, in Figures 5-8 it is worth highlighting the measurement error or at least standard deviations.

Author Response

Reviewer 1

however in Figure 3 the descriptions are illegible (please change the color of the descriptions).

Response 1

Thanks for your suggestions.

I have changed the color of the descriptions in the figure. (page 5 line 112 to 120).

in Figures 5-8 it is worth highlighting the measurement error or at least standard deviations.

Response 2

Thanks for your suggestions.

I have highlighted the standard deviations in figures 5-8. ( page 7 line 159 to 160, page 8 line 172 to line 173, page 9 line 192 to 193).

Reviewer 2 Report

The study conducted by Chengyu Guan and collaborators reports the benefits and drawbacks of employing Sn-based solder alloy on Flip-Chip LED filament. The manuscript is on a topic of relevance for the reader of the Journal, although significant English editing are required. Minor revision have to be considered before the publication.

Please consider to improve the introduction of the manuscript and give the necessary background to the main topic of the research. Other comments in the following:

(1) Abstract: The first time the two bonding materials should be introduced in their general term, following cite the label. Therefore, it is better to use “This article researches the effects of Sn-based solder alloy on flip-chip LED (FC-LED) filament properties.”

(2) Pay attention with the acronym: first report the complete name and then the acronym. For example Light Emitting Diode (LED), Flip-Chip LED (FC-LED).

(3) Explain the connection between “With the growing demand of LED, white LED has been industrialized [4-5]” and the following sentence “FC-LED filament bulbs are gradually replacing […]”. Are FC-LED white emitter? Explain that.

(4) From line 28, the authors state “FC-LED filament bulbs are […] lower working voltage, less power consumption, and higher luminous efficiency.” From line 34, they repeat the same concept: “the FC-LED packaging process has higher luminous efficiency […]”. Avoiding repeating concepts.

(5) Once you introduce FC-LED do not use again flip-chip LED.

(6) Line 37: the sentence “Of which, the reliability of eutectic welding FC-LED filament was higher” is not grammatically correct. Additionally, “higher” is a comparative.

(7) Please void repetition of concepts: for example “heat dissipation” is repeated three times in the last paragraph of the introduction, while “thermal conductivity” four time in the same paragraph.

(8) “Sn90Sb10 (Sn90-Sb10) and SAC0307 (Sn-0.3Ag-0.7Cu) solder WERE selected for the bonding process”.

(9) Line 48: what are SAC solder? Please explain.

Author Response

Reviewer 2

1. Abstract: The first time the two bonding materials should be introduced in their general term, following cite the label. Therefore, it is better to use “This article researches the effects of Sn-based solder alloy on flip-chip LED (FC-LED) filament properties.”

Response 1

Thanks for your suggestions.

I have used “This article researches the effects of Sn-based solder alloy on flip-chip LED (FC-LED) filament properties.” in the abstract. (page 1 line 11 to 12).

2. Pay attention with the acronym: first report the complete name and then the acronym. For example Light Emitting Diode (LED), Flip-Chip LED (FC-LED).

Response 2

Thanks for your suggestions.

I have reported the completed name and then the acronym, and only use FC-LED and LED in the manuscript. (in red).

3. Explain the connection between “With the growing demand of LED, white LED has been industrialized [4-5]” and the following sentence “FC-LED filament bulbs are gradually replacing […]”. Are FC-LED white emitter? Explain that.

Response 3

Thanks for your suggestions

In the process of FC-LED filament package, the FC-LED filament can emit blue light after welding, which can emit white light after baking by coating a proper layer of phosphors on its surface, and the luminous flux ratio is 1:7. The FC-LED filament coated with phosphors is packaged in the bulb, which can be used in various lighting occasions. Compared with the traditional incandescent lamp, this kind of FC-LED filament bulb has many advantages, such as lower working voltage, less power consumption, and higher luminous efficiency, so the FC-LED filament bulb are gradually replacing the incandescent lamp.

4. From line 28, the authors state “FC-LED filament bulbs are […] lower working voltage, less power consumption, and higher luminous efficiency.” From line 34, they repeat the same concept: “the FC-LED packaging process has higher luminous efficiency […]”. Avoiding repeating concepts.

Response 4

Thanks for your suggestions

I have modified the introduction and deleted the repeated concepts. (page 2 line 33 to 59).

5. Once you introduce FC-LED do not use again flip-chip LED.

Response 5

Thanks for your suggestions

I have modified flip-chip LED to FC-LED and only use FC-LED in the manuscript. (in red).

6. Line 37: the sentence “Of which, the reliability of eutectic welding FC-LED filament was higher” is not grammatically correct. Additionally, “higher” is a comparative.

Response 6

Thanks for your suggestions

I have modified this sentence to “This flip-chip LED packaging structure of good heat dissipation property and low stress was the necessary technology. Lead-free solder alloy gradually replaced the conductive silver glue and thermal conductivity adhesive with the advantage of high heat dissipation efficiency and thermal conductivity in the LED wafer packaging field”in the manuscript. (page 2 line 44 to 52).

7. Please void repetition of concepts: for example “heat dissipation” is repeated three times in the last paragraph of the introduction, while “thermal conductivity” four time in the same paragraph.

Response 7

Thanks for your suggestions

I have deleted the repeated concepts “heat dissipation” and “thermal conductivity”. (page 2 line 33 to 59).

8. “Sn90Sb10 (Sn90-Sb10) and SAC0307 (Sn-0.3Ag-0.7Cu) solder WERE selected for the bonding process”.

Response 8

Thanks for your suggestions

I have modified this word “was” to “were”. (page 2 line 53).

9. Line 48: what are SAC solder? Please explain.

Response 9

Thanks for your suggestions

SAC is a Sn-based series of solder pastes used in flip-chip LED packages, and the main components are Sn/Ag/Cu.

I have explain “SAC  solder” in the manuscript. (page 2 line 55 to line 59).

Reviewer 3 Report

No comment as the authors have revised the manuscript accordingly.

Author Response

Reviewer 3

1. No comment as the authors have revised the manuscript accordingly.

Response 1

Thanks for your suggestions

I have revised the manuscript. (in red).

This manuscript is a resubmission of an earlier submission. The following is a list of the peer review reports and author responses from that submission.

Round 1

Reviewer 1 Report

The paper is interesting, however there are some  mistakes, such as:

lack of spaces in whole paper
- in Fig. 1 twice is used (a) and (b)
- Fig. 4 - there is box-plot without description lower and whiskers as well as lower and upper quartile
- there is no discussion

After some corrections, it can be printed. 

Reviewer 2 Report

The manuscript contains the information on the benefits and drawbacks of employing various bonding materials on Flip-Chip LED filament. The authors investigate the optical and electrical properties of Sn99Ag0.3Cu0.7 and Sn90Sb10 lead-free solder welded FC LED filament. In the present form, I see poor significance of the content and scientific relevance. Moreover, the paper requires important editing of English language and style. The topic could be of interest for Journal audience, however it needs to be importantly improved in both description of the method and research design.

In order to help the authors, below a list of some relevant issues that should be addressed before considering it for publication:

(1) Additional comments on the specific choice for the bonding materials should be included (why the authors decide to explore the FC LED properties employing Sn90Sb10 and SAC0307?);

(2) Improve the quality of the figures;

(3) Page 3: improve the experimental description. For all the analyses conducted, the name of the instrument, parameters and experimental conditions have to be reported. For example, which conditions have been used for the SEM images (please, change the verb in the sentence: "microstructures [...] were photographed by SEM").

(4) Page 3, Experimental Section 2: what is 'mil', referring to the FC dimensions?

(5) Page 4: how did you measure the peaking welding temperature? please include a comment on it.

(6) Page 5: the authors use the push pull tester to measure the peak shearing force. Please, insert here the parameters and conditions used for the experiments.

(7) Please make uniform the text: use only Sn90Sb10 and SAC0307, instead of Sn-0.3Ag-0.7Cu

(8) Figure 6 shows the thermal images of the filament with a single temperature: I would suggest to measure the filament temperature for the whole object and showing the average value. Moreover, two figures are sufficient for demonstrating the method (the measurements are quite simple and easy to understand).

(9) Are 15 min sufficient to test bonding material on a FC LED filament? LED can last 50000 hours: is it sufficient to test the variation of luminous flux under ageing for 300 hours?

Reviewer 3 Report

Title is of this manuscript is Effect of different bonding materials on flip-chip LED filament properties. But, their abstract says about optical and electrical properties of....? Doesn't make sense at all. Must be written again.

Introduction is weak.

Figure 2. No scale bar?

Too many figures in the main text.

Conclusion is again weak. Please don't write it in point form.

We are at the end of 2019, yet there is no reference from 2019 can be seen?

Based on the abovementioned issues, I cannot recommend this manuscript for publication.